# Structure of the *Drosophila melanogaster* Flight Muscle Myosin Filament at 4.7 Å Resolution Reveals New Details of Non-Myosin Proteins

**DOI:** 10.3390/ijms241914936

**Published:** 2023-10-05

**Authors:** Fatemeh Abbasi Yeganeh, Hosna Rastegarpouyani, Jiawei Li, Kenneth A. Taylor

**Affiliations:** 1Institute of Molecular Biophysics, Florida State University, Tallahassee, FL 32306-4380, USA; fa16b@fsu.edu (F.A.Y.); hr18d@fsu.edu (H.R.); jli19@fsu.edu (J.L.); 2Department of Biological Science, Florida State University, Tallahassee, FL 32306-4380, USA

**Keywords:** cryo-electron microscopy, myosin, flightin, myofilin, stretchin, striated muscle

## Abstract

Striated muscle thick filaments are composed of myosin II and several non-myosin proteins which define the filament length and modify its function. Myosin II has a globular N-terminal motor domain comprising its catalytic and actin-binding activities and a long α-helical, coiled tail that forms the dense filament backbone. Myosin alone polymerizes into filaments of irregular length, but striated muscle thick filaments have defined lengths that, with thin filaments, define the sarcomere structure. The motor domain structure and function are well understood, but the myosin filament backbone is not. Here we report on the structure of the flight muscle thick filaments from *Drosophila melanogaster* at 4.7 Å resolution, which eliminates previous ambiguities in non-myosin densities. The full proximal S2 region is resolved, as are the connecting densities between the Ig domains of stretchin-klp. The proteins, flightin, and myofilin are resolved in sufficient detail to build an atomic model based on an AlphaFold prediction. Our results suggest a method by which flightin and myofilin cooperate to define the structure of the thick filament and explains a key myosin mutation that affects flightin incorporation. *Drosophila* is a genetic model organism for which our results can define strategies for functional testing.

## 1. Introduction

The striated muscle sarcomere is made up of four essential components: thick, bipolar filaments of myosin; thin, polar filaments containing actin, troponin, and tropomyosin among other proteins; a Z-disk that cross-links antiparallel actin filaments to produce a bipolar structure; and connecting filaments linking thick filaments to the Z-disk. While thin filaments across all organisms are very similar due to the high sequence and structural conservation of actin, thick filament structures are highly variable, especially among invertebrates [1], and in turn are very different in comparison to those of vertebrates. All vertebrates that have been studied extensively have had a single conserved structure [2,3,4] that has recently been exhibited in stunning detail [5,6]. Structural studies of striated muscle thick filaments have numerous challenges, including the high diversity among invertebrates, the presence of only quasi-helical symmetry in vertebrate filaments [7], and challenges in the isolation of filaments for structural studies.

Myosin molecules are heterohexamers containing two pairs of light chains, dubbed essential (ELCs) and regulatory (RLCs), and a pair of heavy chains [8]. The light chains and their bound segment of heavy chain constitute the myosin lever arm. Approximately 850 N-terminal residues and one each of the light chains comprise the head of myosin II, which can be separated from the rest of the molecule via proteolysis in a low-salt buffer dubbed subfragment 1 or S1. Most of the remaining approximately 1150 residues form an α-helical coiled-coil tail. Proteolysis of the tail in a high-salt solution separates it into an N-terminal fragment dubbed subfragment 2 or S2 and a C-terminal fragment dubbed light meromyosin (LMM). The myosin heads of most invertebrate thick filaments that have been studied, are spaced at intervals 145 Å [9]. In *Lethocerus*, the arrangement of myosin heads in relaxed thick filaments give the appearance of a ring encircling the filament backbone, a structure dubbed a “crown”.

The individual myosin heads of an active muscle are independent force generators and are generally disordered except when attached to actin [10]. In relaxed muscle, low calcium concentrations inactivate the thin filament, preventing strong myosin attachments and allowing for easy sarcomere extension [10]. Under relaxing conditions, myosin heads in many striated muscle thick filaments become ordered against the filament backbone as determined via both electron microscopy [11,12,13], X-ray fiber diffraction [11,14] and others. At that time, the myosin head arrangement could not be determined because of the low resolution of the observations.

An unexpected asymmetric head-to-head interaction has been identified in a vertebrate smooth muscle heavy meromyosin fragment (HMM) of myosin II [15], later dubbed the interacting heads motif (IHM). The structure explained much of the actin activated ATPase regulation of smooth muscle HMM [16]. Essentially the same IHM has been seen in the 10S conformation of full-length smooth muscle myosin II [17,18] and now has been visualized at near-atomic resolution [19,20,21].

Originally, the IHM was thought by many to be specific to smooth muscle and vertebrate non-muscle. A subsequent reconstruction obtained from ice-embedded relaxed tarantula striated muscle thick filaments [22] showed that the IHM was not specific to smooth muscle. Subsequent studies over a wide range of isolated myosin II isoforms from different species [23] as well as a wide variety of relaxed thick filaments [3,4,24,25] suggested the structure was almost universal among a wide range of organisms that have a multicellular form [26,27]. The structure is thus highly suited to inhibiting the interaction of myosin heads with actin, so much so that it has been implicated in the so called “super-relaxed” state of striated muscle [28].

In the relaxed thick filaments from most species of striated muscle, the IHM is positioned approximately as first observed in tarantulas, stabilized via charged interactions of the blocked head with the proximal S2 and in some cases interactions between neighboring IHMs [29]. Recent structures of thick filaments from the flight muscles of the large waterbug *Lethocerus indicus* found an unusual orientation, with the IHM lying approximately perpendicular to the thick filament axis [30]. In this orientation, the presumptive blocked head does not interact with the proximal S2 or with neighboring IHMs as is the case in other species. Lacking any interaction between the blocked head and the proximal S2, the IHM binds tangentially to the backbone via the free head. The blocked head is oriented as if to “pin” the free head against the filament backbone.

A recent thick filament reconstruction from the flight muscle of *Drosophila melanogaster* under relaxed conditions showed no ordered myosin heads [31], consistent with early cryoEM results from filaments prepared differently [32]. An RLC mutant that could not be phosphorylated [33] also had disordered heads, thus ruling out phosphorylation as the cause. More recently, a bumble bee thick filament from *Bombus ignitus* was reported to also have disordered heads [34]. It remains unclear whether *Drosophila* flight muscle thick filaments form an IHM that is disordered because it cannot attach to the filament backbone or whether a disordered IHM forms at all.

Among all species, thick filaments have additional proteins that regulate their function or length (Figure 1). The protein obscurin binds to the bare zone of myosin filaments and is long enough to reach the first crown of myosin heads [35]. Obscurin-like proteins are found in both vertebrate and invertebrate thick filaments. The giant protein titin is found exclusively in vertebrate thick filaments. Vertebrate titin performs two functions: it determines the length of the filament [36], and it forms the connecting filament to the Z-disk in which role its stiffness to extension contributes to the muscles ability to be activated by stretch [37]. Two smaller titin-like proteins, projectin and kettin, are part of invertebrate thick filaments [38,39,40] and form the connecting filaments to the Z-disk (Figure 1). Both are found predominantly near the thick filament ends [39]. Stretchin-klp is generally distributed along the A-band in *Drosophila* flight muscle [41]. Four other proteins of invertebrate thick filaments, paramyosin, miniparamyosin [42], flightin, and myofilin [43,44] are in either the filament cores or among the myosin tails (not shown in Figure 1). Vertebrate thick filaments lack these four proteins.

Here we report an improved structure at 4.7 Å resolution of flight muscle thick filaments from a wild type strain of *Drosophila melanogaster*. The improvement over the previous 7 Å resolution structure [31] resolves the α-helix backbone with improved detail, facilitating construction of an atomic model. The reconstruction shows improvement in the head densities, and, though still largely disordered, the proximal S2 region is observed fully for the first time up to the likely position of the head–tail junction. Improved resolution for the non-myosin proteins resolves the entire length of flightin within the annulus of myosin tails, resolves the ambiguity for one unassigned density, and improves the shape of the stretchin-klp densities on the surface of the filament backbone.

## 2. Results

Cryo-EMs of isolated *Drosophila* thick filaments show fine lines forming longitudinal stripes following the filament axis (Figure 2a,b). These lines are most likely the myosin molecule’s α-helical coiled–coiled arrangement, which runs generally parallel to the filament axis [31]. In some locations, myosin layers within the backbone will be oriented edge-on, which would amplify their visibility. No evidence of ordered myosin heads is visible in the motion-corrected images, in contrast to the distinct crown structure seen with *Lethocerus* thick filaments [30]. Original micrographs suggest that the filaments lack significant density in the central core (Figure 2a), following a pattern visualized in multiple fly species using conventional electron microscopy techniques [45].

Using single-particle processing methods, 138,000 filament segments (particles) were picked manually for the reconstruction using the Relion manual picker and extracted to obtain filament segments [46]. These segments were subjected to 2D classification to select 116,000 segments of high quality for the final reconstruction. Generally, the class averages showed only a small density that might correspond to the myosin heads and weak density within the filament core, giving the filament a tubular appearance (Figure 2b). Striking axial density variations are seen, which might represent the non-myosin proteins, flightin, myofilin and stretchin-klp, which follow the myosin helical symmetry but are positioned mostly between crowns [31].

Using 116,000 filament segments, ~1000 Å in length, we computed a 3D reconstruction, using cisTEM [47]. cisTEM estimated the resolution of the reconstruction as 3.85 Å, which corresponds to the highly organized backbone structure. According to the MonoRes map, 4.7 Å is the best resolution (Appendix A). The disordered myosin heads, which are a significant component of the overall mass, reduce the overall resolution.

### 2.1. Myosin Tail Structure and Arrangement

The myosin tail has an approximate length of ~1600 Å. This length encompasses nearly eleven 145 Å axial repeats, or crowns. To visualize a complete myosin molecule in its native context, the reconstruction was extended to a length of 12 crowns using Relion. Individual myosin tails could then be segmented from the map (Figure 3a). At a resolution cutoff of 4.7 Å, approximately 10 crowns of the myosin tail, which represents the part forming the filament backbone, are visible as a continuous density corresponding to individual α-helices. The less ordered proximal S2 is partly visible as the 11th crown at the same density threshold but connects to the disordered myosin head density at a lower threshold (Figure 3b).

The myosin tails are arranged in “curved molecular crystalline layers” [48] which we will refer to as “myosin layers”. Myosin layers consist of tails that are offset axially by three crowns (Figure 3a). Each myosin layer is sandwiched between two other myosin layers, one offset axially by +1 crown, with the other offset by −1 crown (Figure 3c). With rotational and helical symmetry imposed, each myosin layer is identical to every other myosin layer within most of the A-band except for the bare zone boundary and the tapered end, where rotational symmetry is probably maintained, but helical symmetry is not. A transverse slice through a myosin layer (Figure 3c) always cuts through at least three tails but cuts through four tails in one out of 10 crowns. The 11th crown is the proximal S2, which is not in the backbone. Thus, the asymmetric unit, from which most of the entire myosin tail annulus can be assembled via symmetry operations, consists of 10 segments of myosin tails, representing crowns 2–11, from 10 individual myosin molecules. Each axial thick filament repeat consists of four asymmetric units, which rotate by the helical angle as one progresses toward the tapered end.

The helical parameters measured via Relion_helix_toolbox give a helical rise of 145.357 Å, and the helical twist is 33.8076 Å, values which are consistent with X-ray fiber diffraction results [49] and quite like those from *Lethocerus* and *Bombus* (Appendix A).

The myosin tails are arranged around a hollow core with a diameter of 180 Å, unlike the thick filaments of *Lethocerus* and *Bombus*, which contained eight densities believed to correspond to paramyosin [30,34]. Myosin tails in *Drosophila* extend from the N-terminus, located outside the tail annulus, to the C-terminus, located inside the tail annulus, giving an outward tilt of 1.8° relative to the filament axis.

### 2.2. Myosin Heads and Proximal S2 Region

Like the previous *Drosophila* thick filament reconstruction [31], at high resolution, four “floating” densities occur in the expected axial position of myosin heads but are not connected to the visible part of the proximal S2 at 4.7 Å resolution. When the map is low pass filtered to 30 Å resolution the proximal S2 extends to almost the floating densities but the connection is via thin densities that may represent individual α-helices that are part of the proximal S2, but not the coiled coil. Where the proximal S2 exits the backbone, it passes over the Skip 1 region of another myosin tail (Figure 3a). The Skip 1 region of both *Lethocerus* and *Drosophila* has a pair of sharp bends defining the ends of its accommodation region, which is the segment of myosin coiled coil where the two α-helices partially unwind and the coiled coil untwists to accommodate the inserted skip residue [50]. The proximal S2 cannot rejoin the filament backbone after it passes over the accommodation region. When compared to the published 7 Å map [31], nearly all the proximal S2 region in the current map is visible as a coiled coil up to the diffuse density we attribute to the disordered heads (Figure 3c), whereas the 7 Å map showed only a short stubble of proximal S2 where it exits the backbone.

The myosin heads are disordered and are represented in the reconstruction as only a shapeless diffuse density that is centered at the location of the RLCs. Absent F-actin, myosin heads can form two types of structures; they can be either individual myosin heads moving independently, or mobile IHMs moving as one structure. The myosin heads in either conformation are disordered in relaxed *Drosophila* thick filaments and present as a shapeless density (Figure 4a) that does not take on a recognizable shape regardless of the low pass filter level or the density cutoff. When the *Drosophila* thick filament is superimposed on the *Lethocerus* thick filament, which has ordered heads, the two densities, one with shape the other without, superimpose (Figure 4b,c). Using the alignment of the two reconstructions based on their very similar backbone density (Figure 4d,e) shows that both proximal S2s originate at the same axial and azimuthal positions on the backbone. However, the *Drosophila* S2 follows a near-axial path, whereas the similar density in *Lethocerus* is bent by 17° azimuthally (Figure 4d). In a tangential view revealing the radial disposition, both reconstructions are superimposed (Figure 4e). The proximity of the floating density to the beginning of the proximal S2 suggests that the floating density represents disordered myosin RLCs, as previously suggested for the similar density observed in *Bombus* [34].

### 2.3. Homology Model of the Myosin Coiled-Coil

Sharpened maps show the myosin tail α-helical coiled coil in detail (Figure 5). The resolution in the present case is insufficient to resolve all sidechains except for the largest ones, but at 4.7 Å the resolution is high enough to separate the myosin α-helices, observe the coiled–coiled packing and the effects of the four skip residues on the coiled coil.

We used rigid body fitting to compare the *Drosophila* myosin tail density with a *Drosophila* homology model [51], using the *Lethocerus* atomic structure as a template (PDB 7kog). The myosin tail sequence of *Lethocerus* flight muscle is 88% identical and 98% similar to that of *Drosophila*. Therefore, the *Lethocerus* atomic model should be a good template for a homology model for *Drosophila* myosin. The homology model backbone based on *Lethocerus* fit the *Drosophila* reconstruction well as a rigid body fit (Figure 5). The skip residues and their accommodation regions aligned well with the homology based atomic model without further refinement. At Skip 4 (Figure 5i), the density map indicates one α-helix has unfolded.

We used the atomic model to examine charge interactions among the coiled coils. Visualizing the charge interactions in general rather than as specific interactions was a challenge that we think was solved by making a film of the coulomb interactions (Appendix A).

Charge interactions of all types are observed, including ones of opposite and similar charges. The role of the repulsive interactions is likely to make the packing of certain regions looser than those regions where the charge interaction is attractive. A similar film was made illustrating the hydrophobic and hydrophilic surfaces (Appendix A). Most hydrophobic surfaces occur between the two chains of the coiled coil, as predicted for 2-stranded coiled coils [52].

### 2.4. Non-Myosin Proteins

*Drosophila* flight muscle thick filaments have several non-myosin proteins: flightin [53], myofilin [44], stretchin-klp [41,54], paramyosin and miniparamyosin [55], kettin [40], projectin [38], and obscurin [35]. Kettin, projectin and obscurin are either too low in abundance relative to myosin for them to be visible in a helically averaged 3-D reconstruction, or they are located in areas, such as the M-band and the filament ends, that are avoided by segment selection [31]. Proteins, such as paramyosin and miniparamyosin, follow a different symmetry than the myosin molecules and are in low abundance in *Drosophila* compared to *Lethocerus* or *Bombus*. Consequently, they are averaged out in the reconstruction. Using Chimera segmentation, we could separate three non-myosin densities, which we attribute to myofilin, stretchin-klp and flightin (Figure 6).

Myofilin (always shown in yellow) was tentatively identified as the second non-myosin density in *Lethocerus* [30]. The *Drosophila* myofilin amino acid sequence is the smallest of the three species so far reconstructed [34]. Myofilin is predominately found on the inside surface of the myosin tail annulus (Figure 6a–d). The myofilin density in *Drosophila* has a significant contact with a single myosin tail from a single myosin layer. The small domain found in *Drosophila* appears to correspond to the N-terminal domain identified in *Bombus,* based on its shape and superposition with the similar density. Both the *Bombus* and *Drosophila* myofilin have segments that pass between two myosin tails within a myosin layer (Figure 6c). The folded domain of myofilin is close to but does not contact flightin (Figure 6d).

Stretchin-klp (always shown in purple) was first visualized in the 7 Å reconstruction as a chain of three densities repeating along a left-handed helical track on the backbone surface [31]. Two of the densities were well defined, the third much less so. Connectivity was suggested. Here we see the same three density repeat with better defined connections and more density for the third, more heterogeneous density (Figure 6a). The stretchin-klp density is completely on the outside of the filament backbone but is nestled into valleys between myosin tails (Figure 6b). Stretchin-klp makes no visible contact with myofilin (Figure 6a–d). Although stretchin-klp passes over flightin, it makes no visible contact (Figure 6b). Stretchin-klp interacts primarily with a single myosin layer, although its continuity over the five quasi repeats in a single molecule gives it interactions with multiple myosin layers.

Flightin (always shown in red) was first visualized in *Lethocerus* as a small, folded domain, since identified as the WYR domain [34], followed by an extended peptide chain that reaches outside of the myosin tail annulus to contact the proximal S2 [30]. Recently, a flight muscle thick filament reconstruction from *Bombus ignitus* showed that a previously unidentified density, the so-called “blue protein” [30], was connected to the WYR domain via an extended density [34]. An atomic model was built that indicated the flightin C-terminus was located on the inside surface of the myosin tail annulus with a connection to the WYR domain, after which the polypeptide chain was extended to reach outside the backbone. The flightin N-terminus, lacking any stabilizing interaction was not visible in the reconstruction. Non-myosin densities attributable to flightin and myofilin have now been found in *Lethocerus*, *Bombus* and *Drosophila*.

#### 2.4.1. Myofilin

The myofilin density of *Drosophila* is significantly smaller than for *Lethocerus* (Figure 7a–c). Only the globular part at the putative N-terminus has been resolved in *Drosophila*, but it is similar in shape and position to the globular density seen in *Lethocerus* [30] and *Bombus* [34]. This N-terminal globular domain has been dubbed the “LKG” domain in *Bombus* because of its highly conserved sequence pattern along with the similar corresponding densities.

When compared to the myofilin density from *Lethocerus*, *Drosophila* myofilin densities correspond well to similar densities in *Lethocerus*, but *Lethocerus* has additional density in this region that may correspond to some of the 21-residue insertion after *Drosophila* residue I35 that is found only in *Lethocerus* [34]. Much of this 21-residue insertion was visible in *Lethocerus* but not in *Bombus* (Figure 7b,c). Currently, where the rest of the *Drosophila* myofilin sequence is located is unknown.

We modelled the structure of *Drosophila* myofilin using Alphafold. The atomic model of the N-terminus had a helix–loop–helix structure, but the rest of the model bore no resemblance to the observed structure (Appendix A).

#### 2.4.2. Stretchin-Klp

Three additional densities were observed on the outside surface of the *Drosophila* thick filament backbone that were not observed in *Lethocerus* or *Bombus* (Figure 7d,e). Proteomic data indicates that these additional densities correspond to the Ig-like domains and linkers of stretchin-klp [31]. An analysis of the stretchin-klp sequence in UniProt suggested a repeating sequence of Ig-like domains and linkers of various lengths (Table 1). The basic pattern is one of Ig-like (86–92 residues), short linker (12–27 residues), Ig-like (86–90 residues), long linker (61–174 residues). With respect to size, the Ig-like domains and short linkers are more uniform than the long linkers, accounting for the better definition of these densities. The long linkers are more heterogeneous than the short linkers and are generally large enough to form small, folded domains of their own. The structural pattern observed is that of two equally sized domains with a well-defined short linker and a large domain that is relatively poorly ordered, representing the five highly variable long linkers.

The filament segments used in the reconstruction are not necessarily equally distributed with respect to the five repeats. It is therefore possible that the size distribution contributing to the third folded domain corresponding to the long linker is not necessarily random and could conceivably be biased with respect to size in favor of the larger long linkers. Nevertheless, the density corresponding to the long linker is poorly defined.

The atomic structure of an I-set domain of myosin binding protein C (PDB 2YXM; Figure 7d,e) fit satisfactorily according to size into the two globular densities supporting interpretation of these densities. We also modeled one repeat of the “Ig-short linker-Ig-long linker” sequence using Alphafold (Appendix A). The Ig-like domains of this model fit well. The short linker has the same relative length as shown in the density, while the long linker, which was modeled as a long α-helix, was a poor match to the density. The five “long linker” predicted structures were all different. Mostly, they formed folded domain, but in this case, the predicted structure was a single long α-helix. The long linker sequence in the five repeats is highly diverse, which suggests that their structures are highly diverse. All three stretchin-klp densities are positioned near the Skip 1 region of one myosin tail, which lies on the backbone surface.

#### 2.4.3. Flightin

The flightin densities from the three species have a similar shape with three distinctive features (Figure 7f–h). The most prominent is a folded domain approximately in the middle of the segmented density (Figure 7f). Studies of the flightin amino acid sequences from many species identified a highly conserved domain in the middle of the sequence, residues W85-R131 in *Drosophila*, dubbed the “WYR” domain due to its enrichment in tryptophan, tyrosine, and arginine [56]. Extending from this middle domain toward the outside of the myosin tail annulus is a V-shaped extension. The vertex of the “V”, H84, is highly conserved and occurs just before the first residue of the WYR domain. The V-shaped extension ends at the outside surface of the thick filament backbone, but the density continues for a variable number of residues. The atomic model begins at residue P69 leaving 68 N-terminal residues disordered. In *Lethocerus*, the visible part is longer because it is stabilized via contact with the proximal S2, but here in *Drosophila* as well as in *Bombus,* it does not and is consequently shorter. Running along the inside of the myosin tail annulus from the WYR domain is an inner, extended density that connects to the former “blue” density at the C-terminus (Figure 7f, Appendix A). The inner extension is found only in *Drosophila* and *Bombus*; this feature has so far not been seen in *Lethocerus,* although its “blue” density is present.

Application of the structure prediction program AlphaFold [57] to the *Drosophila* flightin sequence produced a structure for the WYR domain that was an excellent fit to the central density. The WYR domain-predicted structure consists of a pair of α-helices connected by a loop. The first helix has a distinct kink in its middle, which allows the two halves to incorporate a bend of ~45° (Figure 7g). The orientation of the predicted WYR domain structure identifies the V-shaped density as coming from the sequence on its N-terminal side and the inner extended density as coming from the C-terminal side of the flightin sequence. To obtain the final model, we removed the N-terminal residues P69-K95 from the AlphaFold model and rebuilt them de novo in COOT. In addition, the residues G140-L182 were also rebuilt de novo to fit the density (Figure 7g). The final flightin atomic model was refined against the reconstruction using the Real Space Refinement utility in Phenix [58] with acceptable validation. There are clashes in the model, especially between the two helical regions but the resolution proved insufficient to define a correction. However, the model in its current state has some predictive power.

For purposes of description, we define the following elements (Figure 7i): external domain (residues A2-G74) [59], extended-1 (residues Y75-R83), extended-2 (residues H84-Y93), helix-1 (residues K94-Q115), loop-1 (residues T116-T127), helix-2 (residues W128-D144), connector (residues S145-Y166), C-terminal (residues N167-L182). The external domain refers to those observed residues that are outside of the thick filament backbone, a major part of which is not visible and is presumably disordered.

Differences in the flightin external domain, which primarily reflect differences in mobility, are seen among the three species. *Lethocerus* flightin has the longest visible external domain, probably stabilized by an interaction with the proximal S2. The visible part of the *Drosophila* external domain is short and contacts nothing. The visible *Bombus* flightin external domain has approximately the same length as that in *Drosophila* but extends outward almost perpendicular to the thick filament surface (Figure 7h).

The extended-1 and -2 elements define the V-shaped loop, which is a structure very similar in shape for all three species we have so far examined. The vertex of the loop in *Drosophila*, residue H84, is highly conserved and occurs just before the beginning of the WYR domain, residue W85, suggesting that it may be the structural beginning of the WYR domain. The beginning of the V-shaped loop, extended-1, is a region of poor sequence conservation among the three species for which structures have been revealed.

Extended-1, helix-1, loop-1 and part of helix-2 are the elements of the WYR domain. The WYR domain technically ends at *Drosophila*, residue R131 after just one turn of helix-2. However, sequence conservation in helix-2 is good up to *Drosophila* residue T135. In comparison to the atomic model for *Bombus* flightin, structure conservation is good up to *Drosophila* residue I138. The helix–loop–helix structure of the WYR motif appears very similar among the three flightin structures that have been observed at subnanometer resolution. The C-terminal end of helix-1 and the beginning of loop-2 may comprise a paramyosin binding site because they extend furthest into the hollow core of the filament.

The flightin connector runs along on the inner side of the myosin tail annulus but makes no stabilizing contacts with the underlying myosin tails. This element, reported first for *Bombus* [34], is absent in *Lethocerus* [30] and was not seen in the previous reconstructions from *Drosophila* [31]. Although the density here is not well defined, given the 22 residues comprising this element and the distance to be spanned, it is likely that most of the *Drosophila* connector structure must be α-helical. The connector comprises a poorly conserved region of sequence and may have comparatively divergent structures. AlphaFold predicted a helical structure for the connector. The 22 residues in the *Drosophila* connector have only a single site of 100% conservation with the other two species and seven sites with 2/3 conservation [34]. *Lethocerus* has the longest connector sequence. *Bombus* has a 5-residue deletion in the sequence and *Drosophila* has a 3-residue deletion. These deletions occur in non-overlapping regions of the connector sequence. Both the poor conservation and shorter amino acid sequence may explain why the connecting densities of *Bombus* and *Drosophila* do not overlap. Thus, the connector may play an important role in defining the structure of the thick filament.

The density at the end of the flightin structure in *Drosophila* and *Bombus* highly overlap with the previously unidentified “blue protein” found in *Lethocerus*, which is strong evidence of its correspondence to part of the flightin molecule even though it failed to connect to the WYR domain. The 16 residues at the end of the flightin C-terminus are highly conserved, with four sites of identical sequence and ten sites with two out of three residues conserved [34]. Sometimes the 3rd residue is a conservative substitution.

A genetic study of *Drosophila* flightin showed that truncation of the 44 residues at the C-terminus (residues 139–182) disrupts flight, although normal sarcomeres were formed at eclosion [60,61]. This region includes all the connector and C-terminal elements. The structures of the putative WYR domain and the C-terminal domain were predicted with higher confidence by AlphaFold than the region at its N-terminal side, which appears to be intrinsically disordered. The connector, which was absent in the *Lethocerus* reconstruction [30], is resolved in *Drosophila* albeit at a lower resolution compared to the rest of the reconstruction.

The atomic models of flightin and myosin showed three sites that appear to be important for incorporation of flightin within the myosin tail annulus (Figure 8a). The myosin mutation E1554K prevents incorporation of flightin into *Drosophila* flight muscle thick filaments [43]. The mutation occurs on the *e* position of the heptad repeat (*abcdef*) [62], which makes it a candidate for interaction with other polyptide chains. The *Drosophila* flightin atomic model places flightin residue R87 in position to form a salt bridge with E1554 (Figure 8b). The same glutamate-to-lysine mutation at the equivalent residue in human β-myosin (MYH7), E1555, has been implicated in human cardiac muscle disease [63].

The second region of interaction (Figure 8c) contains a salt bridge between myosin residue E1557 with flightin loop-1 residue R124. Another salt bridge is found between myosin residue R1857 and flightin helix-2 residue D144 (Figure 8c). These have not previously been reported. An additional region of interaction involves mostly hydrophobic interactions between tyrosines of flightin helix-1 and the myosin tail (Figure 8d). The predicted interactions between the myosin tail and flightin are summarized in Appendix A.

## 3. Discussion

When it comes to protein composition and structure, striated muscle thin filaments are more conserved across species than are thick filaments. Thin filaments across nearly all striated muscle types studied have, at their core, an actin filament with narrowly defined helical parameters in addition to tropomyosin, troponin, and capping proteins at both (+) and (−) ends. The myosin containing thick filaments are highly heterogeneous with regards to structure and composition. Thick filaments from vertebrates are the most uniform, possibly because their length, 1.6 μm, is determined by the giant protein titin [64]. Their rotational symmetry appears uniform at C3 but many lack helical symmetry and even lack a uniform spacing between crowns [3]. Of the thick filament structures so far reported at subnanometer resolution, all have had myosin tails arranged in layers as predicted by Squire [48]. Within the helical segments of invertebrates these myosin tail layers are identical, but recent reports of the myosin tail arrangement of vertebrate thick filaments show three different arrangements of myosin tail layers [5,6] in parallel with the three different head arrangements.

Thick filaments from invertebrates have variable rotational symmetries, with the lowest symmetry being C4. They generally show helical symmetry but vary widely in length and in protein composition. Invertebrate titin-like proteins are much smaller than vertebrate titin, though in some cases their structures are related. For example, the flight muscle proteins kettin and projectin, referred to as minititins, like titin, are composed of chains of Ig and Fn3 domains [65,66]. All invertebrate thick filaments contain the protein paramyosin, whereas no vertebrate thick filament has this protein. These differences may provide keys to understanding what properties of the thick filament proteins lead to different muscle properties.

*Drosophila* also is a genetic model organism with a unique property that specifically aids studies of muscle structure and function; flies can live in the laboratory even if they cannot fly. This property facilitates the formation of homozygous mutant flies which are ideal for structural and functional studies without wild type contamination. Mutations in vertebrate thick filament proteins generally must be heterozygous if the animal is to survive, which will make structural studies more difficult. Studies of thick filaments provide insight into myosin molecule packing and the positions of accessory proteins. Thick filaments of *Drosophila* flight muscle are among the select few, e.g., *Lethocerus*, *Bombus* and tarantula species, for which subnanometer resolution reconstructions have been produced but is the only one which provides the opportunity to understand muscle function and features through mutations. There are a number of mutations in *Drosophila* flight muscles [67] and some similarities between the flight muscle thick filaments of *Drosophila* and *Lethocerus*, but there are also some surprising differences that have been newly revealed through the present high-resolution structure of the *Drosophila* thick filament.

The reconstruction reported here is an improvement over the previous report [31] by a more than 2 Å increase in resolution but, more importantly, by a higher number of filament segments. This has improved the definition of the α-helices, facilitating construction of an atomic model. The accessory proteins, flightin and stretchin-klp, are much more defined, which enabled us to link flightin with a previously unidentified density in the reconstruction.

### 3.1. Disordered Myosin Heads

For the myosin heads in any form to be visualized in an image reconstruction using the methods utilized here, they must be ordered. In *Lethocerus*, the heads are ordered via binding of the free head to the filament backbone and the blocked head binding to the free head. So far, the only thick filament structure from an insect utilizing asynchronous flight muscle with ordered myosin heads has been that from the Hemipteran *Lethocerus indicus* [30]. The procedures that produced ordered heads for myosin thick filaments for *Lethocerus* have not worked for *Drosophila*. This problem dates to the first report of *Drosophila* thick filaments preserved in ice [32]. The only difference in the processing of *Drosophila* thick filament compared to *Lethocerus* was the use of fresh muscle. The possibility of isolation method being responsible for ordering heads, seems unlikely.

The IHM in *Lethocerus* is in a different orientation from other striated muscle thick filaments. The interaction between free and blocked heads within the IHM produces a relatively flat structure, whose position can be approximated by a plane. This plane in most relaxed striated muscle thick filaments is oriented approximately tangential to the thick filament surface, with the blocked head contacting the proximal S2 and with interactions between IHMs along the thick filament helical tracks [22,24]. In *Lethocerus*, this plane is oriented perpendicular to the backbone surface with no interactions between neighboring IHMs and no stabilizing interaction between the blocked head and the proximal S2 [31]. An interaction between the free head and the filament backbone stabilizes the IHM in *Lethocerus*. In *Drosophila*, if an IHM is formed it is disordered, apparently because the presence of stretchin-klp obstructs binding to the filament backbone via the free head [31]. Whether myosin heads form an IHM that is disordered in relaxed *Drosophila* flight muscle is possible, but unlikely because studies on purified myosin from several species found IHM formation to be a general property of most forms of myosin II, but not when isolated from *Drosophila* flight muscle [27].

*Drosophila* may in fact lack the ordered IHM that is present in relaxed muscles of *Lethocerus* and many other striated muscles [30,68]. It is known that RLC phosphorylation disrupts the IHM. A mutant *Drosophila* flight muscle that could not be phosphorylated on the RLC also failed to show ordered heads. *Drosophila* flight muscle also expresses a non-myosin protein, stretchin-klp, which was previously found on the thick filament backbone surface [31,69] in a position to obstruct binding of the free myosin head to the filament surface thereby preventing the ordering of the IHM. Whether or not stretchin-klp prevents IHM formation is a separate question.

### 3.2. Role of the Proximal S2 in Muscle Contraction

The S2 fragment of myosin is thought to function as a tether that facilitates, or restricts, the search for appropriately oriented actin targets for myosin head binding. S2 itself was originally defined as the length of the coiled coil that was present in the HMM subfragment after myosin is digested in high salt by trypsin, ~50 nm [70]. Under such conditions, myosin is monomeric and not filamentous. Many early publications on myosin filament structure depict just such a length for the S2 tether, which exceeds an axial distance of three crowns.

Few attempts have been made to measure the functional length of the S2 tether, which can be determined by swelling the rigor filament lattice in low ionic strength solutions and viewing the connections between myosin heads and the thick filament via thin section electron microscopy. It can also be determined directly by a high-resolution image reconstruction of relaxed thick filaments from different species.

One such study on *Lethocerus* flight muscle found that only ~10 nm of S2 could be pulled free of the thick filament backbone [71]. That number was later confirmed via a subnanometer image reconstruction of flight muscle thick filaments isolated *Lethocerus* [30], *Drosophila* and *Bombus* thick filament reconstructions also showed that 11 nm of S2 was free of packing constraints within the backbone [31,34]. Previous reconstructions of *Drosophila* thick filaments showed just a short stub of S2 because the myosin heads themselves were disordered [31], but the location of that stub agreed with the *Lethocerus* result so it can reasonably be assumed that the length of the S2 tether in all three flight muscle thick filaments is also 11 nm. Thus, in *Lethocerus*, *Drosophila* and *Bombus*, the length of the myosin tether is short of one crown in length, considerably less than depicted in early descriptions. Similar experiments have not been done on other muscles to our knowledge so it cannot be said with certainty that the length of the S2 tether in other muscles is shorter than the ~50 nm length defined by proteolysis in high salt.

Just how much movement radially and axially is allowed by an 11 nm length of S2 depends on S2’s persistence length. A number of investigators have studied the persistence length of α-helical coiled coils, but this discussion is based on that of Hvidt et al. [72]. They found a persistence length at 20 °C for the myosin tail of ~130 nm, which is 80% of the coiled-coil length. Using the following equation $${\left(y^{2}\right)}^{1/2}={\left(\frac{L^{3}}{3q}\right)}^{1/2}$$where *L* is taken to be the unrestrained length of the S2 tether, 11 nm, and *q* is the persistence length, 130 nm, we can solve for the root mean square displacement expected at the unrestrained end of the S2.

The result, 1.85 nm, is just about the width of a coiled coil. Since our specimens were frozen at 4 °C, the expectation is that the root mean square displacement will be proportionally less, perhaps 7% less given the ratios of the temperatures. Thus, the ability to resolve the S2 tether when the heads are disordered seems within expectations based on the measured persistence length of the myosin coiled coil.

In the first 3-D image reconstruction of the *Drosophila* flight muscle thick filament, only a short stub of the proximal S2 was visible; the rest was disordered [31]. As such, no apparent limitation on the amount of movement of the myosin heads could be estimated. In the present reconstruction, even though the heads are also disordered, resulting in a similar floating density at the expected axial position of the myosin heads, nearly the entire S2 tether is visible up to where the *Lethocerus* coiled coil began. Although this connection is only visible after low-pass filtering to lower resolution, the fact that it is visible puts some limitation on the amount of movement of the myosin heads. The S2 connection to the floating myosin head density occurs on its perimeter nearest the thick filament backbone. The floating density itself is asymmetric having an ellipsoidal shape with the shortest axis being along the direction of the filament axis, suggesting that the dominant movement of the myosin heads is azimuthal in direction. A spherical floating density would imply similar movements in all directions. One could conclude from this that the myosin heads have a similar range of movement at the end of the S2 tether as they would have had were the head-tail junction being held against the thick filament backbone. The proximal S2 mostly moves the pivot point of the head movements roughly 25 Å away from the thick filament backbone.

In rigor, *Lethocerus* flight muscle at least one head of every myosin is able to attach to a thin filament [73], whereas the bending stiffness of the proximal S2 would imply that only a limited number of myosin heads can find binding sites on actin. In flight muscle, active contractions occur on the order of milliseconds. For example, the wing beat frequency of *Lethocerus* is ~40 Hz or one wing beat every 25 msec. For *Drosophila*, the contractions are even faster at 250 Hz [49] or about one wing beat every 4 msec. Rigor contractures are generally much slower and once the cross bridge is formed, it remains attached to actin [74]. Thus, rigor formation can lead to many more cross bridge attachments than might occur during active contraction. It is worth pointing out that in *Lethocerus*, both rigor and active myosin head attachments occur in the target zone, which comprises the four actin subunits midway between troponin complexes. However, other rigor attachments can occur near the troponin complex [75].

## 4. Conclusions

Model organisms such as *Drosophila* are essential for studying muscle. Understanding the structural features will provide insights into thick filament development and its function within the sarcomere.

Myosin layers, which are structural building blocks of thick filaments, and actin, structural building blocks of thin filaments, exhibit remarkable similarity in different orders of IFM. F-actin structure is highly conserved across most striated muscles. The minor differences are among F-actin binding proteins such as troponin and tropomyosin, which lead to different functional features [76]. *Drosophila* is a member of the Diptera insect order that appeared ~245 million years ago, while *Lethocerus* is member of the Hemiptera insect order that appeared ~373 million years ago [77]. It is only natural that there would be distinct differences in the thick filaments of their flight muscles since the Hemiptera diverged from Diptera so long ago. In the myosin layers, the myosin tail structure and arrangement are nearly identical, suggesting that evolutionary pressure has been exerted toward maintaining the myosin tails. Improved structure shows more distinct structural differences between two species. In addition to this, *Drosophila* is a unique model system for analyzing cardiomyopathy mutations and their effects on the wings and the sarcomeres. While invertebrates existed long before vertebrates, there is a high level of sequence identity in the myosin tail between different species of *Placopecten magellanicus* (scallop) and *Drosophila*. In many different kinds of striated muscles, the amino acid sequences of the myosin tails are conserved, especially the charged residues necessary for filament formation [78]. This suggests that the myosin layers that form the myosin filaments may be very similar, even though their placement within the vertebrate thick filament backbone differs compared with invertebrates.

## 5. Materials and Methods

### 5.1. Thick Filament Preparation

Myosin filaments were isolated from the wild type strain W1118 of the fruit fly *Drosophila melanogaster* under relaxed conditions as previously described [31]. To produce a high yield of well-preserved filaments, enzymatic dissociation of the flight muscle with calpain-1 was carried out, and for more sample purification, thin filaments were removed by fragmentation with calcium insensitive human plasma gelsolin.

### 5.2. Electron Microscopy

Isolated thick filaments were applied to a copper Quantifoil R2/1 reticulated carbon grid as previously described [31]. The grids were then plunged into liquid ethane using a Vitrobot Mark IV (Thermo Fisher Scientific, Waltham, MA, USA) and screened on a Titan Krios G1 at Florida State University operated at 300 keV. Grids were shipped to Pacific Northwest Center for Cryo-EM (PNCC), where data were collected using fringe-free imaging on their Titan Krios G2 #4 equipped with a Gatan K3 mounted at the end of a BioQuantum energy filter, which removed inelastically scattered electrons. A total of 14,501 films (projections) were collected at a magnification of 64,000×, with a pixel size of 1.33 Å/pixel. Films were recorded with a total dose of ~55 $e^{-}/{Å}^{2}$ and defocus range of −0.5 to −2 μm.

### 5.3. Data Analysis

The individual films were frame-aligned with damage compensation using Motioncor2 [79] and CTF correction using GCTF [80] to produce the projections used for the reconstruction. We manually picked filaments using the Relion manual picker and used Relion [81] helical extraction to obtain filament segments. We picked around 138,000 segments with a box size of 768 × 768 pixels, containing approximately six axial repeats of length 145 Å. We “polished” the segments with three rounds of 2D classification to remove the bad segments and false positives using cisTEM [47]; in each round, the best classes that showed details of thick filaments were chosen (Figure 2b) leaving ~116,000 segments for 3D refinement and reconstruction. We performed an ab initio reconstruction using cryoSPARC [82] and obtained an initial reference which was used for 3D refinement in cisTEM. After two rounds of both auto and manual refinement, we used GCTF to estimated localized CTF parameters of the segments. Refining particles with the updated CTF parameters in cisTEM improved the resolution by 1 Å.

Local Monores [83] was used to estimate the local resolution (Appendix A), and Local Deblur [84] to sharpen the map; both programs were utilized within the Scipion framework [85]. Sharpening masks were produced in Relion, which is an essential component to improve sharpening quality. We used relion_helix_toolbox within Relion3 to determine the helical rise and twist. It yielded a rise of 145.41 Å a helical twist of 33.81°. We then imposed helical symmetry on the reconstructed thick filament and extended its length to 12 crowns. Map segmentation was carried out using Chimera [86].

The flightin atomic model started with the AlphaFold prediction model as the initial structure, for which only the WYR domain was a good fit. The poorly fitted regions on either side of the WYR domain were built de novo in Coot [87], and then the whole model was refined in Phenix [88] to achieve the best validation score. The myosin atomic model was firstly built on the *Lethocerus* thick filament (PDB 7KOG) [50]. Given the 90% identity of tail sequence between *Lethocerus* and *Drosophila*, a homology model was built by mutating the *Lethocerus* sequence to *Drosophila* in Coot, and then the model was fitted into Drosophila density and refined in Phenix.

## Figures and Tables

**Figure 1 ijms-24-14936-f001:**
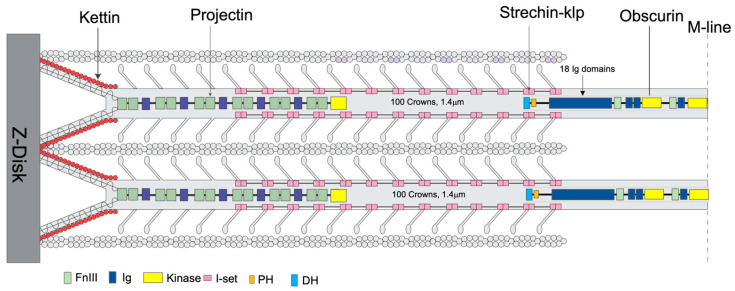
Location of some non-myosin proteins. An illustration of the location of kettin, obscurin, projectin, and stretchin-klp within a sarcomere. The filament tip is mainly bound to projectin, obscurin to the bare zone (M-band), and kettin to the thin filaments and projectin. The stretchin-klp protein binds along the main shaft of thick filaments but not at the bare zones or filament tips. Coloring scheme—Fn3 domains are light green; Ig domains, blue; Ig domains of stretchin-klp, pink; kettin, red; kinase domains, yellow; and DH-PH domains of obscurin, light blue and orange respectively. Adapted from [31,35].

**Figure 2 ijms-24-14936-f002:**
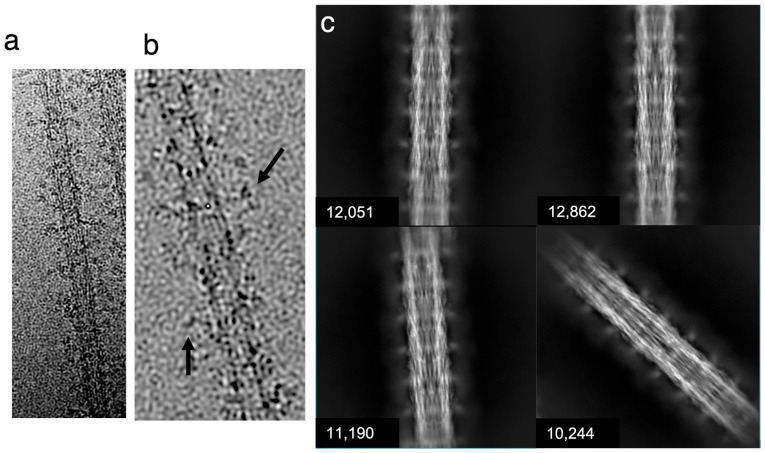
CryoEM of isolated *Drosophila* thick filaments. (**a**) Example micrograph showing the comparatively “hollow” center and disordered myosin heads. (**b**) A higher magnification image of the filament. Dark transverse densities (black arrows) are possibly the myosin heads. (**c**) Four of the 2D classes generated via cisTEM, showing the number of segments (particles) in each selected class average. These classes were used along with others for the final reconstruction. The class averages show a small, periodic density separated from the thick filament backbone presumably representing the average position of disordered myosin heads and emphasizing the hollow nature of the thick filament.

**Figure 3 ijms-24-14936-f003:**
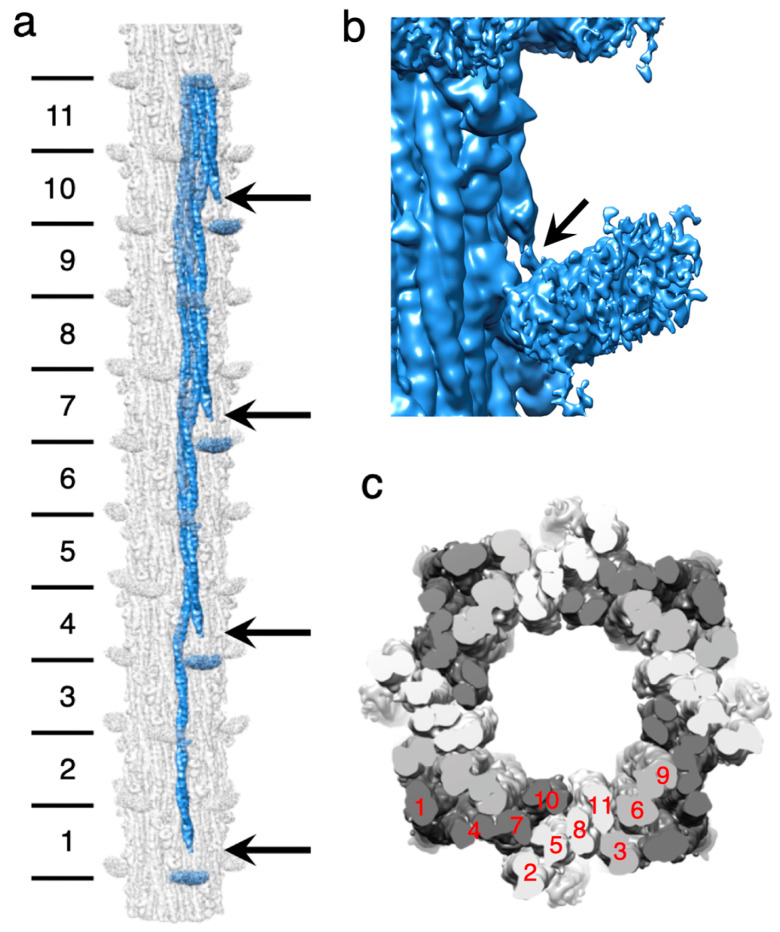
*Drosophila* thick filament reconstruction. (**a**) The 12-crown pseudofilament. The panel illustrates the building of one myosin layer (blue) by successive addition of myosin tails with offsets of three crowns. Lines on the left mark the individual segments of the myosin tail that define the crown spacing. They are numbered starting at the N-terminus. Crown 1 of the coiled coil is slightly shorter than the other 10, whose length is determined by the 145 Å axial spacing. Crown-1 constitutes the proximal S2, which is not embedded in the backbone and is only partially visible in the figure. The floating densities, which represent the averaged density of the disordered heads, approach but do not connect to the myosin tail at the highest resolution. (**b**) In a low pass filtered map (see main text) most of the proximal S2 is visible. Arrows mark the beginning of the visible coiled coil. Adjacent myosin tails of the myosin layers are spaced three crowns apart. Just above and below the arrows, a pair of sharp azimuthal bends occur in the coiled coil, which define the Skip 1 region (arrow) near the floating density. The short stub of the proximal S2 passes over the second (upper) bend of the Skip 1 region. (**c**) Transverse slice through the thick filament with the three myosin layers colored white, light gray, and dark gray. The individual segments are numbered (in red) according to their position in panel (**a**). The dark gray myosin layer shows three pieces of myosin tail plus some of the proximal S2; the white myosin layer shows four pieces within the backbone and the light gray layer shows three pieces.

**Figure 4 ijms-24-14936-f004:**
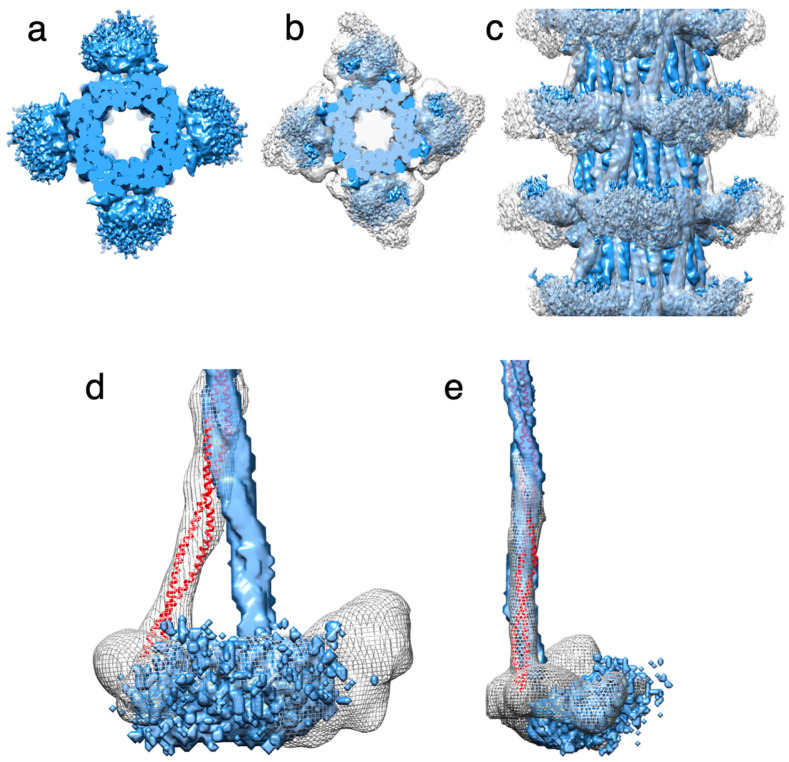
Arrangement of heads and S2 Densities in the *Drosophila* thick filament. (**a**) Transverse view of the low pass filtered structure of *Drosophila* thick filament (blue). (**b**) Reconstruction from panel A superimposed on the *Lethocerus* thick filament reconstruction (white transparent map), showing the relative position of heads which superimpose quite well. (**c**) Side view of panel (**b**). (**d**) Radial view showing the *Drosophila* and *Lethocerus* proximal S2 region with different azimuthal angles. The *Lethocerus* proximal S2 is shown as a red ribbon, and the envelope shown as a mesh. The *Lethocerus* Proximal S2 region is angled 17° relative to the filament axis enforced by the binding of the free head to the myosin tail annulus. (**e**) Side view of panel (**d**). The blue envelope is the *Drosophila* coiled coil, the red ribbon is the *Lethocerus* proximal S2 region, gray mesh is *Lethocerus* S2 + head density. The proximal S2 region of both *Lethocerus* and *Drosophila* are superimposed in this view with only a small outward angle.

**Figure 5 ijms-24-14936-f005:**
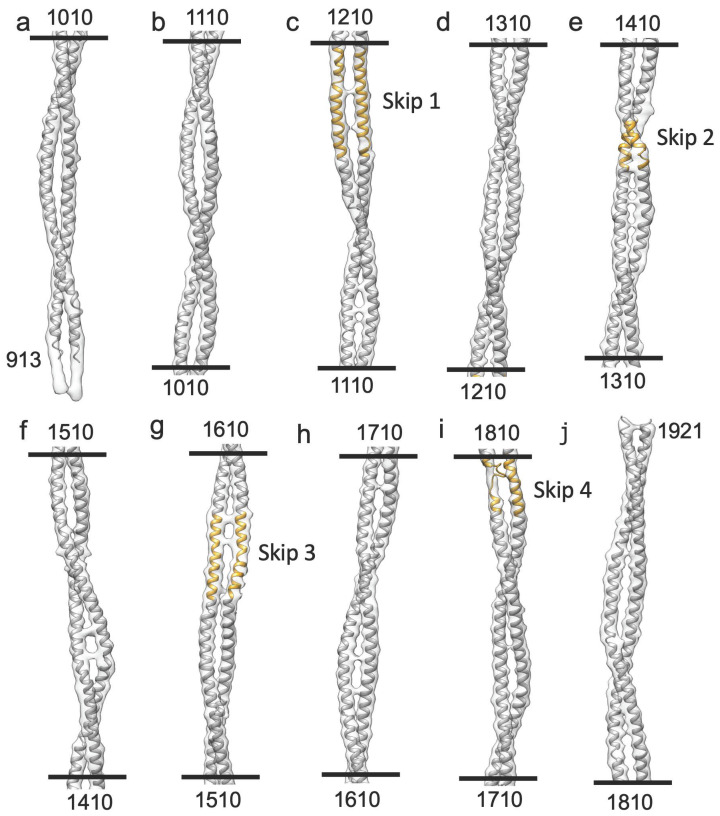
The *Drosophila* homology model of the myosin tail. The entire myosin tail is shown in 10 segments, each with a length of one crown (100 residues) (**a**–**j**). The homology model of the tail starts at residue number 913 and terminates at residue 1921 (The proximal S2 region is not defined at this contour). Numbers at the top and bottom of each segment indicate the starting and ending residue numbers, respectively. The skip residue regions are colored in gold and show a good fit with the density. (**c**) Skip 1, where the density makes a distinct azimuthal shift. (**e**) Skip 2, where the density reveals a normal coiled-coil twist. (**g**) Skip 3, where one chain shows a distinct bend where flightin passes between two myosin tails. (**i**) Skip 4, where the thin density on one chain indicates that the α-helix has unfolded.

**Figure 6 ijms-24-14936-f006:**
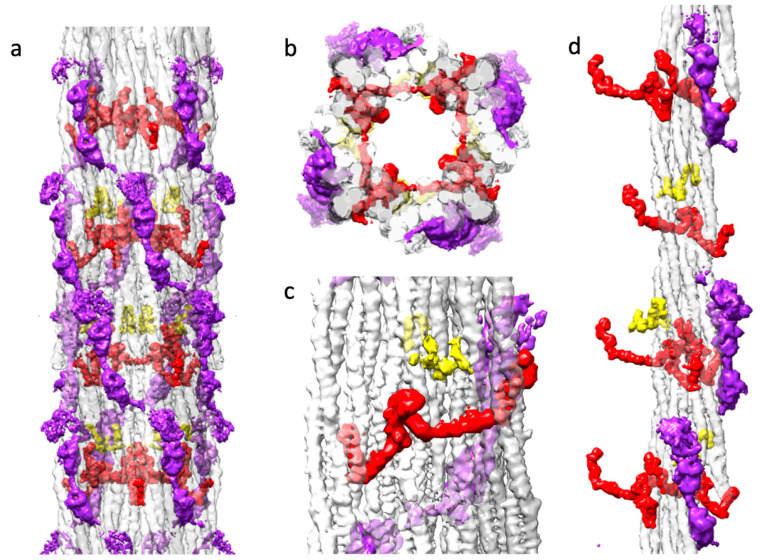
Three non-myosin densities of the *Drosophila* thick filament. Coloring scheme: flightin (red), myofilin (yellow), stretchin-klp (purple). (**a**) The distribution of three non-myosin proteins in the context of the myosin tails. Flightin and myofilin are mostly within the myosin tail annulus. Stretchin-klp is mostly on the outside. (**b**) The top view showing C4 symmetry of *Drosophila* thick filament with all non-myosin proteins. The myosin layer assembly scheme is illustrated by transparent white, light gray and dark gray colors, as used in Figure 3c. Flightin (red), myofilin (yellow), strechin-klp (purple). Note that stretchin-klp domains insert relatively deeply into the myosin tail annulus. The WRY domain of flightin bulges into the empty core. (**c**) View from the inside looking out showing the flightin and myofilin relationship to the myosin layers. Both flightin and myofilin bind the surface of the myosin layers and both pass through at least one myosin layer between a pair of coiled coils. (**d**) The distribution of three non-myosin proteins in the context of a single myosin layer.

**Figure 7 ijms-24-14936-f007:**
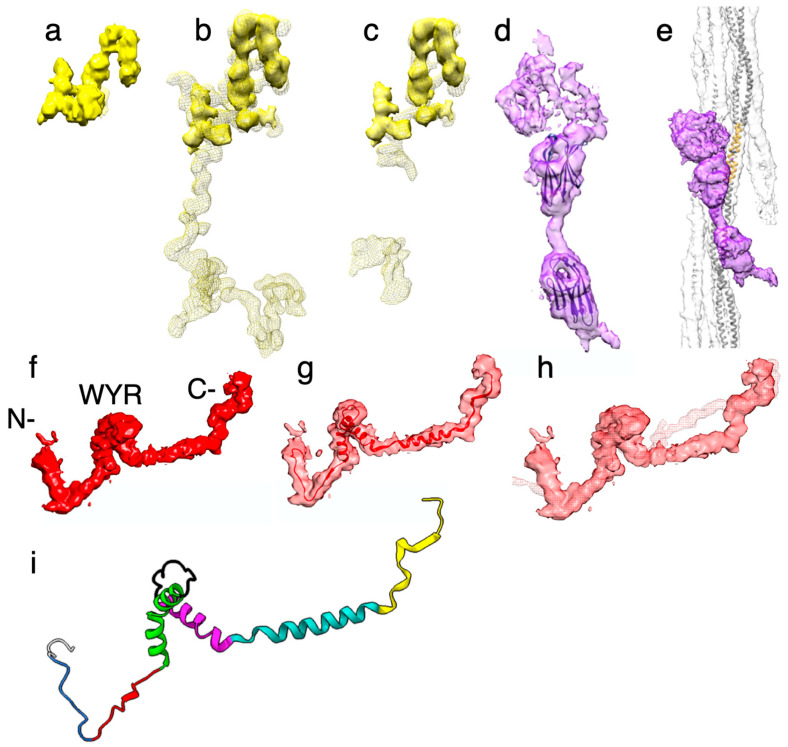
Individual non-myosin protein densities. (**a**–**c**) Myofilin densities from *Drosophila* (yellow solid surface), *Lethocerus* (yellow mesh) and *Bombus* (yellow mesh). (**d**,**e**) stretchin-klp (purple). (**f**–**h**) Flightin. (**a**) *Drosophila* myofilin. (**b**) *Drosophila* myofilin superimposed on the *Lethocerus* myofilin density. (**c**) *Drosophila* myofilin superimposed on *Bombus* myofilin. (**d**) The three densities of the strechin-klp pseudorepeat with an I-set domain atomic model superimposed on the pair of putative I-set domains. The short linker separates them. The long linker at the top is heterogeneous and poorly represented. (**e**) Stretchin-klp density superimposed on a myosin layer. (**f**) Segmented *Drosophila* flightin density (red solid surface). The N- and C-termini are marked. (**g**) *Drosophila* flightin atomic model (red ribbon) superimposed on its transparent (red) density. (**h**) Flightin from *Drosophila* (red transparent surface) superimposed on *Bombus* flightin (red mesh). Density from most of the external domain is disordered. (**i**) The different regions of the flightin atomic model. External domain (white), extended-1 (royal blue), extended-2 (red), Helix-1 (lime green), Loop-1 (black), Helix-2 (magenta), Connector (light sea green), C-terminal (yellow).

**Figure 8 ijms-24-14936-f008:**
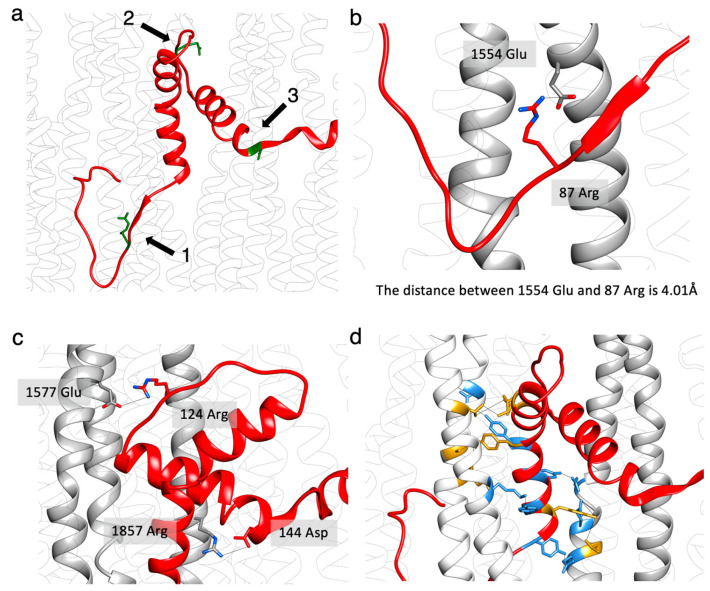
Flightin–myosin interactions**.** (**a**) Three sites along the flightin “WYR” domain form apparent salt bridges with the surrounding myosin, indicating potential roles in anchoring flightin in place and stabilizing the interaction between the WRY domain and myosin. (**b**) The first site is on the left side of the “WYR” domain and along the extended-1 segment of the “V” shape. R87 (positively charged) of flightin with E1554 (negatively charged) of myosin forms a salt bridge. Residue E1554 has been shown to be a key site for flightin incorporation as well as the binding site for Myosin-binding protein C and myomesin in vertebrate myosin. The mutation of residue E1554K results in the absent of flightin in the thick filament [43]. The change of glutamic acid to lysine will break the interaction and inhibit flightin accumulation. (**c**) The second and third salt bridges are roughly in the middle and right side of the “WYR” domain with flightin R124 and myosin E1577 forming one salt bridge and flightin D144 and myosin R1857 forming another. (**d**) There are many potential tyrosine interactions between the “WYR” domain and myosin tails involving hydrogen bonds (colored blue) or hydrophobic interactions (colored gold) on the left-side interface.

**Table 1 ijms-24-14936-t001:** Domain structure of stretchin-KLP.

Domain	Residue Range	Domain Name	Domain Length	Linker Length
1	456–543	Ig-like	88	11
2	554–643	Ig-like	90	67
3	710–798	Ig-like	89	17
4	815–904	Ig-like	89	61
5	965–1056	Ig-like	92	12
6	1068–1153	Ig-like	86	93
7	1246–1333	Ig-like	88	12
8	1345–1434	Ig-like	90	174
9	1608–1693	Ig-like	86	27
10	1720–1808	Ig-like	89	110

From (https://www.uniprot.org/uniprotkb/A1ZA73/entry (accessed on 1 Feburary 2023)).

## Data Availability

The full reconstruction of the *Drosophila* thick filament as well as the segmented flightin and myosin density have been deposited in EMDB under accession code EMD-42024. The atomic model of *Drosophila* flightin is deposited in the PDB under the entry 8U8H, and myosin is deposited with entry 8U95.

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
