# Peer review of "Structure of the Drosophila melanogaster Flight Muscle Myosin Filament at 4.7 Å Resolution Reveals New Details of Non-Myosin Proteins"

_ijms, 2023, doi:10.3390/ijms241914936_

Round 1

Reviewer 1 Report

The authors described the structure of the Drosophila melanogaster flight muscle myosin and its associated non-myosin proteins, flightln.

This is a well -written paper containing interesting results which merit publication. A few minor revisions are needed and listed bellow.

What is the advantage to use Drosphola frying muscle? Filamentaous actin or myosin filaments are larger than other animals, such as rat or mouse skeletal muscle? Or, authors concentrate flightln, one of the novel myofibrillar proteins in this manuscript?   Please cleary indicate major concerns using Drosophila fright muscles (espetially flightlin proteins). 

In Figure 2 (a). CryoEM image is too small to judge the author’s opinion. Higher magnification is needed. For the reader’s convenience, it is recommended conventional TEM image of Drosophila muscle should be included in Figure 2, if possible.

More detailed explanation of “Materials and Methods” are needed. Especially electron microscopy methods and data Analysis. I think 3D images are reconstructed using electron microscopy images (micrograph), it is unclear to understand how to perform 3D reconstruction. Units of ångström (

 Å ) is familiar with this field? If so, it is OK.

I think the English language is enough quality.

Reviewer 2 Report

This manuscript employs Cyro-EM and AlphaFold to analyze the structure of Myosin filament backbone. The manuscript is well-organized, with high quality of figures.  If you can describe the difference between the structure of myosin filament backbone from Fly and that from Mammals in your Introduction, it will be better and benefits more readers. Moreover, I also notice that your group’s paper published Life Sci Alliance (Life Sci Alliance 2020, 3, (8), e202000823) also report the structure of thick filaments of the flight muscle of the fruit fly. Could you clarify the novelty of this work compared with your previous one in the Discussion?

Some minor concerns:

1.     For Figure 1, this illustration is the same with your previous paper (Life Sci Alliance 2020, 3, (8), e202000823). For the citation of line 93 of page 3, “Adapted from [32]” should be “Adapted from [33]”. Please make sure that your manuscript has correct citations in the rest of parts.

2.     For Figure 2, please indicate the meanings of the number “12051” “12864” “11190” “10244”. Figure 5 also has the same problems.

3.     For Figure 8, does E1554K mutation relate with human diseases? Please give some background of this mutation.

4.     For line 537 of page 15, “The mutation occurs on the e position of the heptad repeat [63] which makes it a candidate for interaction with other polyptide chains.”  changed into “The mutation occurs on the position of the heptad repeat [63] which makes it a candidate for interaction with other polyptide chains.”

Minor editing of English language
